# The Sec1/Munc18 protein Vps45 holds the Qa-SNARE Tlg2 in an open conformation

**Travis J Eisemann, Frederick Allen[†], Kelly Lau[†], Gregory R Shimamura[†], Philip D Jeffrey, Frederick M Hughson\***

Department of Molecular Biology, Princeton University, Princeton, United States

**Abstract** Fusion of intracellular trafficking vesicles is mediated by the assembly of SNARE proteins into membrane-bridging complexes. SNARE-mediated membrane fusion requires Sec1/Munc18-family (SM) proteins, SNARE chaperones that can function as templates to catalyze SNARE complex assembly. Paradoxically, the SM protein Munc18-1 traps the Qa-SNARE protein syntaxin-1 in an autoinhibited closed conformation. Here we present the structure of a second SM–Qa-SNARE complex, Vps45–Tlg2. Strikingly, Vps45 holds Tlg2 in an open conformation, with its SNARE motif disengaged from its Habc domain and its linker region unfolded. The domain 3a helical hairpin of Vps45 is unfurled, exposing the presumptive R-SNARE binding site to allow template complex formation. Although Tlg2 has a pronounced tendency to form homo-tetramers, Vps45 can rescue Tlg2 tetramers into stoichiometric Vps45–Tlg2 complexes. Our findings demonstrate that SM proteins can engage Qa-SNAREs using at least two different modes, one in which the SNARE is closed and one in which it is open.

**\*For correspondence:**
hughson@princeton.edu

[†]These authors contributed equally to this work

**Competing interests:** The authors declare that no competing interests exist.

## Introduction

Membrane trafficking in eukaryotic cells is mediated by vesicles that transport cargo from one intracellular compartment to another or to the plasma membrane for exocytosis. Cargo delivery requires that the vesicle and target membranes fuse in a process mediated by SNARE proteins (*Südhof and Rothman, 2009*). Most SNAREs are tail-anchored membrane proteins, and they drive fusion by forming membrane-bridging complexes that draw the vesicle and target membranes into close apposition. Essential for this process are conserved regions about 65 residues in length called SNARE motifs, which in most SNAREs are immediately adjacent to a C-terminal transmembrane anchor. Four complementary SNARE motifs—one each from the R-, Qa-, Qb-, and Qc-SNARE groups—zipper together to form a parallel α-helical bundle (*Fasshauer et al., 1998*; *Kloepper et al., 2007*; *Sutton et al., 1998*). SNARE motifs are grouped according to the zero-layer residue in the middle of the motif, either arginine (R) or glutamine (Q). The four zero-layer residues interact within the otherwise hydrophobic core of the assembled SNARE bundle to help prevent out-of-register assembly of the four α-helices. The remaining core residues within the SNARE bundle are denoted by layer numbers ranging from approximately –8 at the N-terminal end of the SNARE motif to +8 at the C-terminal end.

In addition to complementary SNAREs embedded in the vesicle and target membranes, fusion requires Sec1/Munc18-family (SM) proteins (*Baker and Hughson, 2016*). SM proteins function as SNARE chaperones, capable of both inhibiting and catalyzing the assembly of fusogenic SNARE complexes. The first crystal structure of an SM–SNARE complex, the neuronal SM protein Munc18-1 (hereafter called Munc18) bound to the cytoplasmic portion of the Qa-SNARE syntaxin-1 (Stx), was particularly influential (*Misura et al., 2000*). It showed that an SM protein could function as a clamp to keep the bound SNARE in an autoinhibited closed conformation. In this closed conformation, the

three-helix bundle Habc domain of Stx folds back onto the SNARE motif, preventing the SNARE motif from entering into a SNARE complex. The closed conformation of the Qa-SNARE is wedged into a large cleft in the SM protein and requires the intervention of a third protein—Munc13-1—for opening (*Ma et al., 2011*). Finally, the extreme N-terminal region of Stx, termed the N-peptide, binds to a distinct site on the outside of the SM protein (*Burkhardt et al., 2008*). Munc18 thus makes direct contact with three different regions—the N-peptide, the Habc domain, and the SNARE motif—distributed along the entire length of the Stx cytoplasmic domain.

Two recent crystal structures imply that SM proteins, instead of or in addition to functioning as Qa-SNARE clamps, can act as templates to initiate SNARE complex assembly (*Baker et al., 2015*). These structures showed that the SM protein Vps33 binds the SNARE motifs of the Qa-SNARE Vam3 and the R-SNARE Nyv1 at adjacent sites. Combining these crystal structures yielded a model of the Vps33–Vam3–Nyv1 template complex. In it, the two SNARE motifs adopt a half-zippered configuration, in which their N-terminal halves up to the zero layer are aligned for assembly. Single-molecule force microscopy studies confirmed that the SM proteins Vps33, Munc18, and Munc18-3 all form template complexes with their cognate R- and Qa-SNAREs, and that these template complexes likely represent rate-limiting intermediates in SM-catalyzed SNARE complex assembly (*Jiao et al., 2018*).

Here, we present X-ray crystal structures of the SM protein Vps45, both alone and in complex with the Qa-SNARE Tlg2 (*Cowles et al., 1994*; *Nichols et al., 1998*). Vps45, Tlg2 and its cognate SNAREs, and the CATCHR-family tethering complex GARP all function in trafficking from the endosome to the trans-Golgi network (*Abeliovich et al., 1998*; *Conibear and Stevens, 2000*; *Holthuis et al., 1998*; *Pérez-Victoria and Bonifacino, 2009*; *Siniossoglou and Pelham, 2001*; *Siniossoglou and Pelham, 2002*). In the crystal structure of the Vps45–Tlg2 complex, Vps45 makes direct contact with the N-peptide, Habc domain, and SNARE motif of Tlg2. In contrast with the Munc18–Stx complex, however, the bound Qa-SNARE adopts an open conformation. Specifically, the Habc does not pack against the N-terminal region of the SNARE motif and instead appears to leave it free to initiate SNARE complex assembly. Thus, Qa-SNARE clamping may be a specialized property of Munc18, rather than a general property shared broadly among SM proteins. Tlg2, but not Vps45–Tlg2, is prone to homo-oligomerization. Taken together, our results underscore the ability of SM proteins to prevent SNARE misassembly and template proper assembly.

## Results

### Crystal structure of Vps45–Tlg2

There are four families of SM protein—Sec1/Munc18, Sly1, Vps33, and Vps45—and every eukaryote has at least one member of each family (*Kloepper et al., 2007*; *Koumandou et al., 2007*). Crystal structures have been reported for representatives of the Munc18, Sly1, and Vps33 families (*Baker et al., 2013*; *Baker et al., 2015*; *Bracher et al., 2000*; *Bracher and Weissenhorn, 2002*; *Burkhardt et al., 2008*; *Burkhardt et al., 2011*; *Colbert et al., 2013*; *Graham et al., 2013*; *Hackmann et al., 2013*; *Hu et al., 2011*; *Hu et al., 2007*; *Misura et al., 2000*). We began, therefore, by determining the crystal structure of a member of the fourth family, Vps45 from the thermotolerant fungus *Chaetomium thermophilum*, and refining it to 2.0 Å resolution (*Table 1*). Like the other structurally characterized SM proteins, Vps45 exhibits a three-domain architecture with a large cleft between domains 1 and 3a (*Figure 1A*). Three surface regions are particularly well conserved among members of the Vps45 family, presumably because they are functionally important (*Figure 1—figure supplement 1A*). The first conserved region corresponds to the walls of the cleft. The second and third conserved surface regions correspond to the N-peptide and R-SNARE binding sites observed in previous structures (*Baker et al., 2015*; *Bracher and Weissenhorn, 2002*; *Burkhardt et al., 2008*; *Hu et al., 2011*; *Hu et al., 2007*). Thus, each of the surface regions known to engage SNARE proteins in one or more of the other SM families is highly conserved in the Vps45 family.

We next sought to determine the structure of Vps45 in complex with its cognate Qa-SNARE, Tlg2 (*Bryant and James, 2001*; *Burkhardt et al., 2008*; *Carpp et al., 2006*; *Dulubova et al., 2002*). Bacterial co-expression of *C. thermophilum* Vps45 and *C. thermophilum* Tlg2 residues 1–310, including all but the C-terminal 17 residues of the Tlg2 cytoplasmic domain, yielded a stable 1:1 complex. Small crystals were obtained that diffracted to about 3.5 Å resolution. Better crystals, diffracting to

**Table 1.** Data Collection and Refinement Statistics.
Values in parenthesis correspond to the highest-resolution shell.

| | Vps45 | Vps45–Tlg2$_{1-310}$ | Vps45–Tlg2 | Vps45$_{V306D,F335R}$–Tlg2 |
|---|---|---|---|---|
| **Data collection** | | | | |
| Beamline | CHESS (F1) | NSLSII (FMX) | NSLSII (FMX) | NSLSII (FMX) |
| Wavelength (Å) | 0.9782 | 0.9794 | 0.9794 | 0.9793 |
| Space group | P2$_1$2$_1$2$_1$ | P2$_1$22$_1$ | P2$_1$2$_1$2$_1$ | P2$_1$ |
| Cell dimensions | | | | |
| a, b, c (Å) | 62.49, 93.96, 102.62 | 58.38, 89.43, 209.14 | 58.73, 180.06, 202.09 | 89.14, 58.79, 191.18 |
| α, β, γ (°) | 90.0, 90.0, 90.0 | 90.0, 90.0, 90.0 | 90.0, 90.0, 90.0 | 90.0, 97.75, 90.0 |
| Resolution (Å) | 35–2.00 (2.03–2.00) | 29–3.88 (4.30–3.88) | 30–2.80 (2.88–2.80) | 30–5.12 (5.73–5.12) |
| Completeness (%) | 99.6 (98.9) | 99.3 (98.1) | 99.8 (98.9) | 98.0 (56.4) |
| Redundancy | 4.6 (4.2) | 13.0 (13.4) | 13.4 (12.3) | 6.2 (5.9) |
| R$_{merge}$ | 0.053 (0.564) | 0.246 (2.149) | 0.134 (1.880) | 0.305 (1.222) |
| R$_{meas}$ | 0.092 (0.807) | 0.256 (2.234) | 0.144 (1.961) | 0.334 (1.344) |
| $<I/\sigma_I>$ | 11.7 (2.0) | 6.9 (1.3) | 13.6 (1.4) | 3.3 (1.1) |
| CC$_{1/2}$ | 0.940 (0.743) | 0.993 (0.649) | 0.999 (0.731) | 0.980 (0.564) |
| *Refinement* | | | | |
| Resolution (Å) | 35–2.00 (2.04–2.00) | 30–3.90 (4.29–3.90) | 30–2.80 (2.85–2.80) | |
| No. reflections | | | | |
| Work | 43025 | 9900 | 51270 | |
| Free | 2268 | 505 | 2676 | |
| R$_{work}$ | 0.178 (0.262) | 0.191 (0.291) | 0.194 (0.291) | |
| R$_{free}$ | 0.218 (0.300) | 0.242 (0.382) | 0.248 (0.326) | |
| No. atoms | 4551 | 5707 | 11550 | |
| Average B-factor (Å$^2$) | 42.5 | 212.8 | 93.4 | |
| RMSD | | | | |
| Bond lengths (Å) | 0.007 | 0.005 | 0.008 | |
| Bond angles (°) | 0.795 | 0.8 | 0.814 | |
| Ramachandran | | | | |
| Favored (%) | 98.5 | 94.4 | 95.5 | |
| Outliers (%) | 0.6 | 1.4 | 1.0 | |
| PDB Code | 6XJL | 6XMD | 6XM1 | |

2.8 Å resolution, were obtained after deleting Tlg2 residues 201–228, which are missing from the otherwise nearly identical *Chaetomium globosum* Tlg2 (*Figure 1—figure supplement 1B*). The deleted segment is located near the beginning of the 58-residue linker that connects the Habc domain and the SNARE motif. The structures of the Vps45–Tlg2$_{1-310}$ and Vps45–Tlg2$_{1-310, Δ201-228}$ complexes were determined by molecular replacement with Vps45 as a search model (*Table 1*). Because there were no significant differences between the two structures, we focus hereafter on the higher resolution Vps45–Tlg2$_{1-310, Δ201-228}$ (henceforth simply Vps45–Tlg2) structure (*Figure 1B*). All of the well-characterized regions of Tlg2 are visible and interact directly with Vps45: the N-peptide, the Habc domain, and the SNARE motif. The sequences connecting these regions, as well as the C-terminus of the SNARE motif, appear to be disordered based on the lack of interpretable electron density. The corresponding regions were likewise disordered in the Munc18–Stx complex, with the important exception of the linker between the Habc domain and the SNARE motif, which will be discussed below. Comparing the Vps45 and Vps45–Tlg2 structures reveals that Vps45 does not undergo a major conformational change when it binds to Tlg2. The cleft does, however, open up

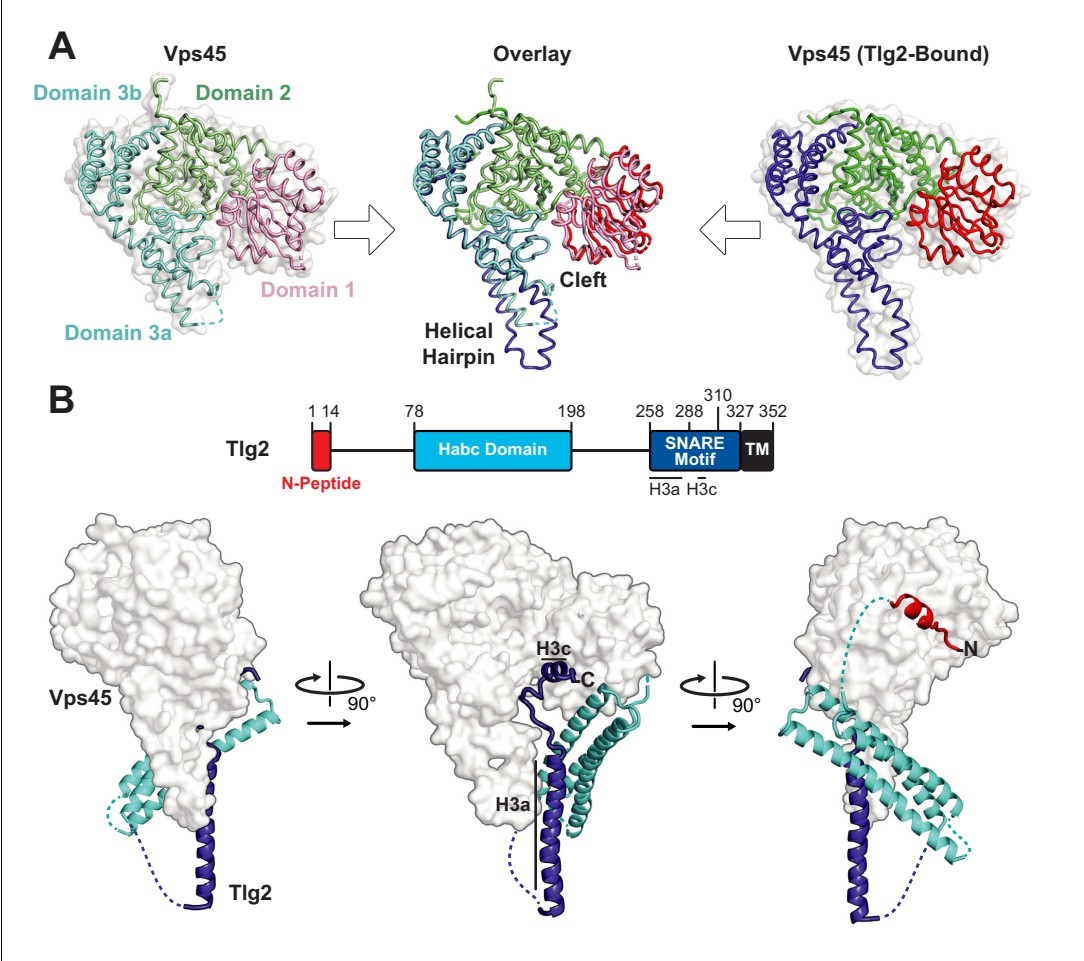

**Figure 1.** Crystal structures of *C. thermophilum* Vps45 and Vps45–Tlg2. (**A**) Crystal structures of Vps45 (left) and Vps45–Tlg2 (right, showing only Vps45). The comparison shown in the center was generated by aligning domains 2 and 3b. (**B**) Crystal structure of Vps45–Tlg2. H3a and H3c are helical regions within the SNARE motif that were defined by Misura et al. based on the Munc18–Stx structure (*Misura et al., 2000*).

The online version of this article includes the following figure supplement(s) for figure 1:

**Figure supplement 1.** Sequence conservation in Vps45 and Tlg2.

slightly, via small rotations of domains 1 and 3a relative to the rest of Vps45, in order to accommodate Tlg2 (*Figure 1A*).

## Bound Tlg2 adopts a novel open conformation

The N-peptide of Tlg2 (residues 1–14) binds to the outside (as opposed to the cleft side) of Vps45 domain 1 (*Figure 2A*). The N-peptide binding mode of Vps45, in which residues 6–12 form a short α-helix, resembles that observed previously for the Munc18 and Sly1 families of SM protein. The interaction between the Tlg2 N-peptide and Vps45 buries 870 Å$^2$ of surface accessible area. Arg residues at Tlg2 positions 3, 5, and 13 appear to play especially important roles, each of them forming both H-bonds and salt bridges with Vps45, while Tyr 9 is completely buried in a deep pocket in which its hydroxyl group forms a charge-stabilized H-bond. Previous site-directed mutagenesis supports the importance of Arg 3, Tyr 9 (often Phe), and Arg 13 for high-affinity Vps45–Tlg2 binding (*Burkhardt et al., 2008*; *Carpp et al., 2006*; *Dulubova et al., 2002*). Near the other end of the Tlg2 polypeptide chain, layers 0 to +4 of the SNARE motif interact with the opposite, cleft-facing side of Vps45 domain 1 (*Figure 2B*). This interaction buries about 930 Å$^2$ surface accessible area. As previously observed in the Munc18–Stx complex (and also in the complex between Vps33 and the Vam3 SNARE motif), the +2 and +3 layers of the Qa-SNARE form a short α-helix (H3c), with the +2 layer

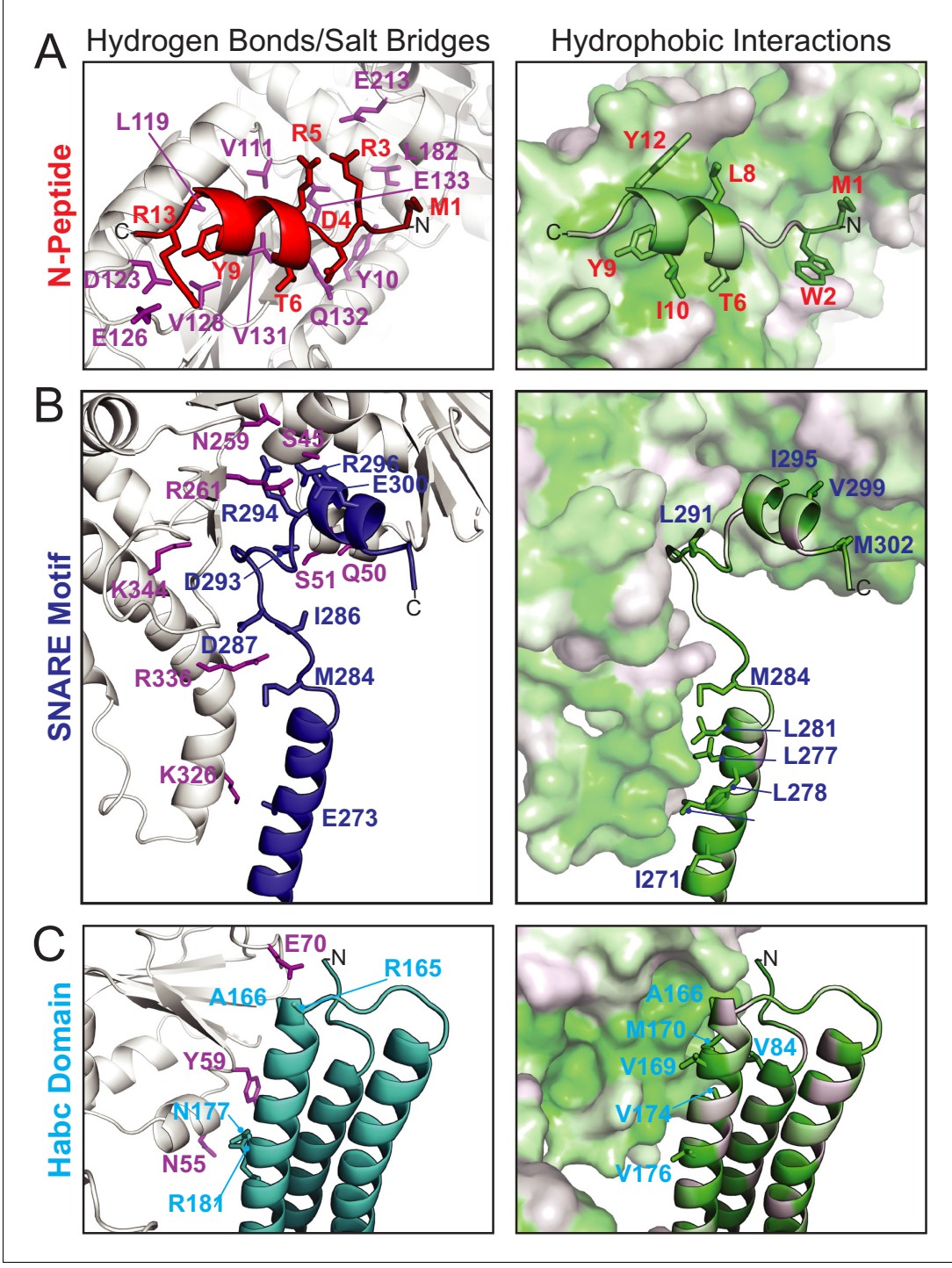

**Figure 2.** Vps45–Tlg2 interactions. (A–C) Interactions between Vps45 and the (A) N-peptide, (B) SNARE motif, and (C) Habc domain of Tlg2. The left panels depict residues contributing to hydrogen bonds and/or salt bridges, with Vps45 residues colored purple. The right panels depict hydrophobic interactions, with residues colored from least (white) to most (green) hydrophobic.

Ile (residue 295 in Tlg2) fitting snugly into a deep hydrophobic pocket on domain 1 (*Baker et al., 2015*; *Misura et al., 2000*). In sum, two short regions at opposite ends of Tlg2, the N-peptide and the H3c helix, bind to opposite sides of Vps45 domain 1 in a manner that closely resembles the Munc18–Stx complex.

By contrast, the intervening regions of Vps45—comprising the Habc domain, the linker that connects Habc and the SNARE motif, and the N-terminal half of the SNARE motif—adopt a novel open conformation. *Figure 3A* compares Sso1 (a yeast exocytic Qa-SNARE that, by itself, is tightly closed [*Munson et al., 2000*; *Nicholson et al., 1998*]), Stx (from the Munc18–Stx complex), and Tlg2 (from the Vps45–Tlg2 complex). Instead of folding with the SNARE motif to form a four-helix bundle-like structure, the Habc domain of Tlg2 makes a limited contact with the SNARE motif, centered around the −2 layer of the latter and at an approximately 45° angle. As noted above, there is no

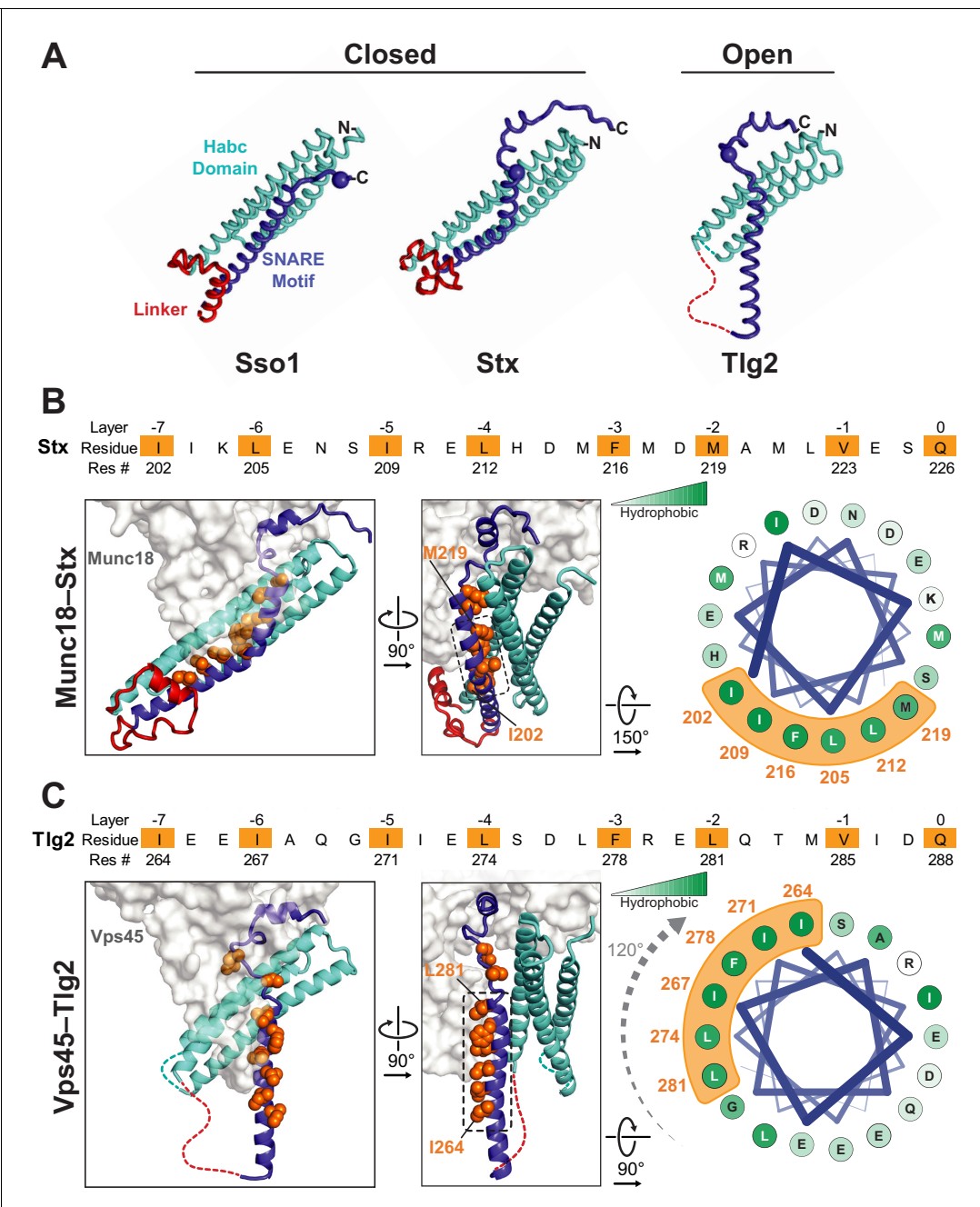

**Figure 3.** Vps45-bound Tlg2 adopts an open conformation. (A) Comparison of uncomplexed Sso1 (PDB code 1FIO), Munc18-bound Stx (3C98), and Vps45-bound Tlg2. The locations of the zero-layer Gln residues are indicated with spheres. (B and C) Comparison of the Habc domains and SNARE motifs in the Munc18–Stx and Vps45–Tlg2 complexes. The core residues of the SNARE motifs (layers −7 to 0) are depicted as orange spheres. At the right, helical wheel representations depict the relative rotation of the SNARE motifs (layers −7 to −2, corresponding to helix H3a in *Figure 1B*) with respect to the rest of the structure.

interpretable electron density for the linker connecting the Habc domain and the SNARE motif, which in the Munc18–Stx complex forms a short helix that packs at right angles against the closed Habc-SNARE motif four-helix bundle (*Figure 3A*). This is particularly notable inasmuch as this region has long been recognized as critical for Stx opening and the initiation of SNARE assembly (*Dulubova et al., 1999*; *Lai et al., 2017*; *Ma et al., 2011*; *Misura et al., 2000*; *Munson et al., 2000*; *Wang et al., 2017*; *Yang et al., 2015*).

The Habc domain interacts with the cleft side of Vps45 domain 1, burying 680 Å$^2$ surface accessible area in an interaction that is largely hydrophobic in nature (*Figure 2C*). Compared to the Munc18–Stx complex, the C-terminal end of the Habc domain has swung away from the SM protein, rotating approximately 15° around an axis near the N-terminal end of the domain (*Figure 3B,C*). As a consequence, the Habc domain does not make contact with domain 3a. Finally, the N-terminal half of the SNARE motif, in lieu of bundling with the Habc domain, forms a helix (H3a) that engages in a limited interaction with the fully extended domain 3a helical hairpin of Vps45 (*Figure 3C*). This interaction buries a surface accessible area of about 300 Å$^2$, with the −4 and −5 layer residues of H3a packing against the long helix of the hairpin. As a result, H3a is rotated approximately 120° around its long axis relative to the Munc18–Stx complex (right-hand panels of *Figure 3B,C*). Overall, the Vps45–Tlg2 structure establishes that SM proteins can engage the N-peptide, the Habc domain, and the SNARE motif of a cognate Qa-SNARE without simultaneously clamping that SNARE in a closed conformation.

## The Vps45 helical hairpin is unfurled

The domain 3a helical hairpin (*Figure 1A*) plays a key role in SM protein function (*Baker et al., 2015*; *Boyd et al., 2008*; *Hu et al., 2011*; *Jiao et al., 2018*; *Parisotto et al., 2014*; *Sitarska et al., 2017*; *Wang et al., 2020*). In Vps45 alone, the distal tip of the helical hairpin is disordered (*Figure 1A*). In the Munc18–Stx complex, it adopts a furled conformation in which the distal tip folds back on the more proximal portion, concealing the R-SNARE binding site (*Misura et al., 2000*; *Figure 4A,B*). The bound Stx, moreover, sterically blocks unfurling (*Baker and Hughson, 2016*; *Hu et al., 2011*). In the structures of Vps33 bound to the SNARE motifs of Vam3 and Nyv1, by contrast, the helical hairpin adopts an unfurled, extended conformation (*Figure 4B*) that interacts extensively with the Qa- and R-SNARE respectively (*Baker et al., 2015*; *Figure 4C*). An unfurled conformation is also seen in the Vps45–Tlg2 complex (*Figures 1A* and *4*), made possible by the open conformation of the bound Qa-SNARE (*Figure 3C*). The strong similarity between the unfurled Vps33 and Vps45 hairpins (*Figure 4B*) suggests that the unfurled conformation may be a well-defined active conformation. Unfortunately, we were unable to identify in vitro conditions under which *C. thermophilum* Tlg2, with or without Vps45, assembles into SNARE complexes. Nonetheless, based on the crystal structure, the Vps45–Tlg2 complex appears to be primed to bind an R-SNARE and, as demonstrated for SM proteins of the Sec1/Munc18 and Vps33 families (*Jiao et al., 2018*), to catalyze SNARE assembly (*Figure 4C*).

A caveat to the above conclusions is that, as observed previously in other SM protein structures (*Baker et al., 2015*; *Hu et al., 2011*; *Wang et al., 2020*), the domain 3a helical hairpin in the Vps45–Tlg2 structure participates in crystal contacts that could potentially influence its conformation. Specifically, the hairpin interacts with the Tlg2 SNARE motif (layers –8 to –4) of a crystallographically adjacent Vps45–Tlg2 complex (*Figure 4—figure supplement 1A–D*). To examine the potential influence of this crystal contact on the Vps45–Tlg2 structure, we sought to disrupt it via site-directed mutagenesis. The hairpin mutant Vps45$_{V306D,F335R}$ formed stable complexes with Tlg2 that crystallized in a new space group (*Table 1*). Although these crystals diffracted only to about 5 Å resolution, electron density maps phased by molecular replacement with a model of Vps45–Tlg2 lacking the SNARE motif allowed us to reach two unambiguous conclusions. First, layers –8 to –4 of the Tlg2 SNARE motif no longer interact with the helical hairpin, instead curving away to make a minor crystallographic contact with the SNARE motif of a neighboring Tlg2 molecule (*Figure 4—figure supplement 1E*). Second, while electron density at this resolution cannot reveal subtle differences, the remainder of the structure—including the position of the Habc domain and the unfurled helical hairpin—appears to be unchanged.

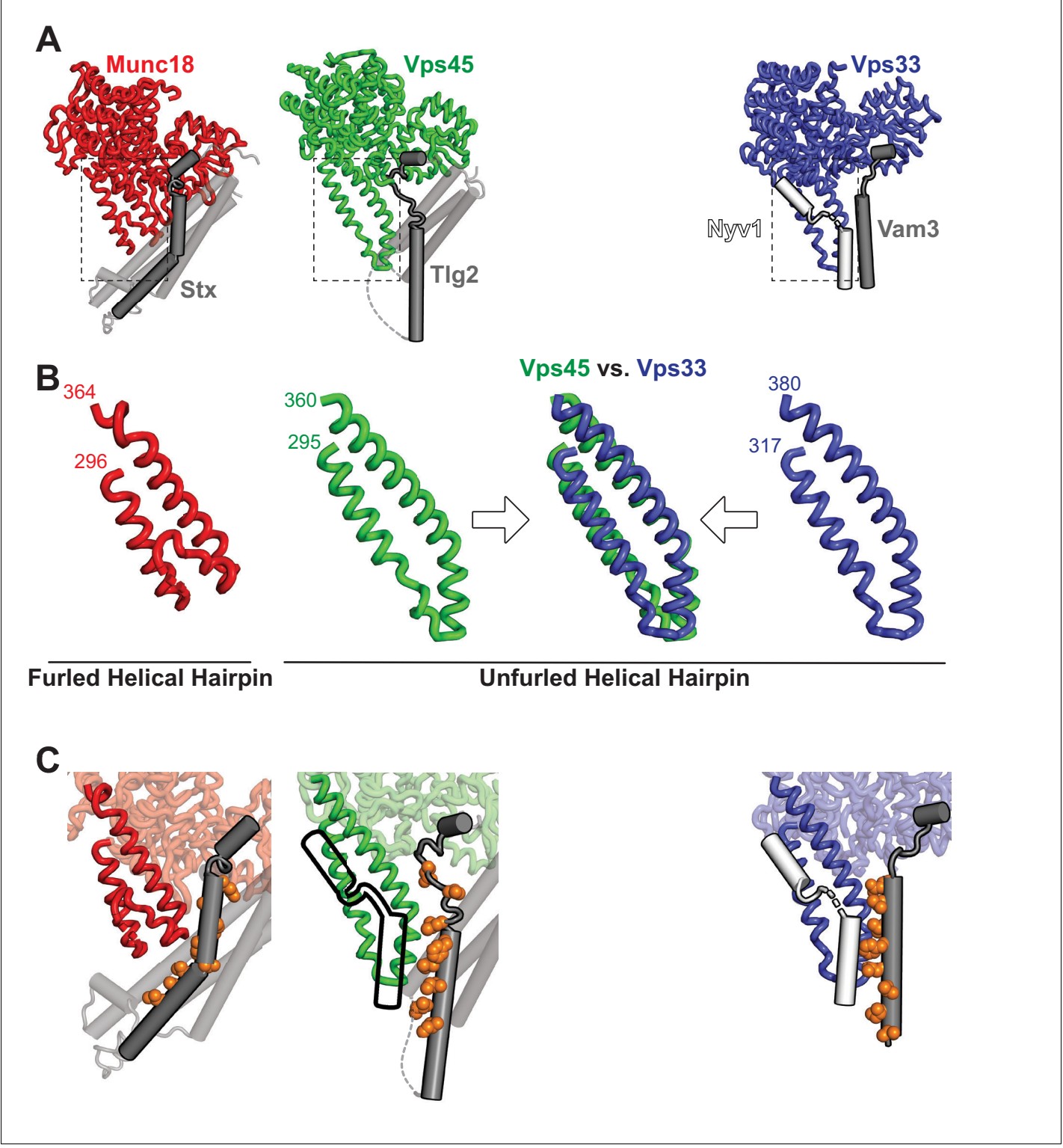

**Figure 4.** The domain 3a helical hairpin of Tlg2-bound Vps45 is unfurled. (**A**) Structures of SM–Qa-SNARE complexes Munc18–Stx and Vps45–Tlg2, as well as a model of the SM–Qa-SNARE–R-SNARE template complex Vps33–Vam3–Nyv1 (***Baker et al., 2015***). The Qa-SNARE motifs are dark gray, the remainder of the Qa-SNAREs are light gray, and the R-SNARE Nyv1 is white. The model of the template complex was obtained by combining Vps33 and Vam3 from the Vps33–Vps16–Vam3 structure (5BUZ) with Nyv1 from the Vps33–Vps16–Nyv1 structure (5BV0); both of these structures lack SNARE N-terminal regions and contain only the SNARE motifs. (**B**) Close-up views of the domain 3a helical hairpins (dashed rectangles in panel A). (**C**) SM–

*Figure 4 continued on next page*

*Figure 4 continued*

SNARE interactions, emphasizing the domain 3a helical hairpins and the SNARE motifs. For Vps45, a model of the template complex is suggested by superimposing the outline of Vps33-bound Nyv1. As in *Figure 3*, the core residues of the SNARE motifs (layers −7 to 0) are shown as orange spheres. The online version of this article includes the following figure supplement(s) for figure 4:

**Figure supplement 1.** Crystal contacts between the domain 3a helical hairpin and the SNARE motif.

## Vps45 prevents Tlg2 oligomerization

Although *C. thermophilum* Vps45–Tlg2 forms a 1:1 complex, the maltose binding protein tagged full-length cytoplasmic region of Tlg2 (MBP-Tlg2$_{1-327}$) formed oligomers as judged by size exclusion chromatography (*Figure 5A,B*). The Tlg2 SNARE motif (MBP-Tlg2$_{258-327}$) likewise formed oligomers, whereas the MBP-tagged SNARE motifs from the cognate R-, Qb-, and Qc-SNAREs (Snc2, Vti1, and Tlg1) did not (*Figure 5C*). These observations suggested that Tlg2 oligomerization, like Stx oligomerization (*Lerman et al., 2000*; *Misura et al., 2001*), is driven by the formation of SNARE complex-like bundles of SNARE motifs. Consistent with this hypothesis, sedimentation velocity analytical ultracentrifugation analysis of both MBP-tagged and untagged Tlg2$_{258-327}$ showed that the Tlg2 SNARE motif behaves as a single species with an apparent molecular mass indicative of a tetramer (*Figure 5D,F* and *Figure 5—figure supplement 1*).

To test whether oligomerization is reversible, we mixed MBP-Tlg2$_{1-327}$ with an equimolar amount of Vps45. After a 2 hr incubation, the majority of the Vps45 shifted into two higher molecular weight peaks, one of them likely representing Vps45 bound to MBP-Tlg2$_{1-327}$ oligomers, the other representing the 1:1 Vps45–MBP-Tlg2$_{1-327}$ complex (*Figure 6A*). At longer incubation times, the 1:1 complex peak became more prominent at the expense of the peak representing Vps45 bound to MBP-Tlg2$_{1-327}$ oligomers, and the Vps45-only peak disappeared altogether. These data imply that Vps45 is able to rescue MBP-Tlg2$_{1-327}$ from the oligomeric state, presumably by trapping MBP-Tlg2$_{1-327}$ monomers that transiently dissociate (*Figure 6B*). As expected, the N-peptide (MBP-Tlg2$_{1-20}$) bound to Vps45 (*Figure 6—figure supplement 1A*). The N-peptide:Vps45 interaction appears to underlie the ability of Vps45 to rescue Tlg2 from oligomers, since N-terminally truncated Tlg2 constructs (MBP-Tlg2$_{21-327}$ and MBP-Tlg2$_{258-327}$) formed oligomers that neither bound to nor were rescued by Vps45 (*Figure 6—figure supplement 1B,C*).

## Discussion

For two decades, the only reported structures of a full-length SNARE cytoplasmic domain bound to an SM protein have been those of the mammalian Munc18–Stx complex (*Burkhardt et al., 2008*; *Misura et al., 2000*) and the highly similar Munc18–Stx complex from the choanoflagellate *Monosiga brevicollis* (*Burkhardt et al., 2011*). These structures have played a pivotal role in the development of models for SM protein function and mechanism. The structure reported here, of the almost full-length cytoplasmic domain of Tlg2 bound to Vps45, is strikingly different. Rather than a four-helix-bundle-like closed conformation, the bound SNARE adopts a much more open conformation, with only a glancing interaction between the Habc domain and the SNARE motif (*Figure 3A*). The linker between the Habc domain and the SNARE motif, which in the Munc18–Stx complex is a target for Munc13's Stx-opening activity (*Dulubova et al., 1999*; *Lai et al., 2017*; *Ma et al., 2011*; *Misura et al., 2000*; *Wang et al., 2017*; *Wang et al., 2020*; *Yang et al., 2015*), appears to be disordered. Finally, the domain 3a helical hairpin, which plays a crucial role in SNARE templating (*Baker et al., 2015*; *Boyd et al., 2008*; *Hu et al., 2011*; *Jiao et al., 2018*; *Parisotto et al., 2014*; *Sitarska et al., 2017*), is furled in the Munc18–Stx complex but unfurled—and therefore primed for R-SNARE binding—in the Vps45–Tlg2 complex (*Figure 4*).

There have been conflicting reports regarding the tendency of yeast Tlg2 to oligomerize and/or form a closed conformation (*Dulubova et al., 2002*; *Furgason et al., 2009*; *Struthers et al., 2009*). The SNARE motif of *C. thermophilum* Tlg2, however, unambiguously forms tetramers (*Figure 5D,F*). In this respect it resembles the SNARE motif of Stx which, by forming four-helix bundles, can drive the formation of off-pathway products requiring rescue by NSF/α-SNAP and Munc18 (*Ma et al., 2013*). In both instances, binding to the cognate SM protein prevents SNARE oligomerization. For Munc18–Stx, this protective effect has been attributed to the closed conformation of Stx. In the

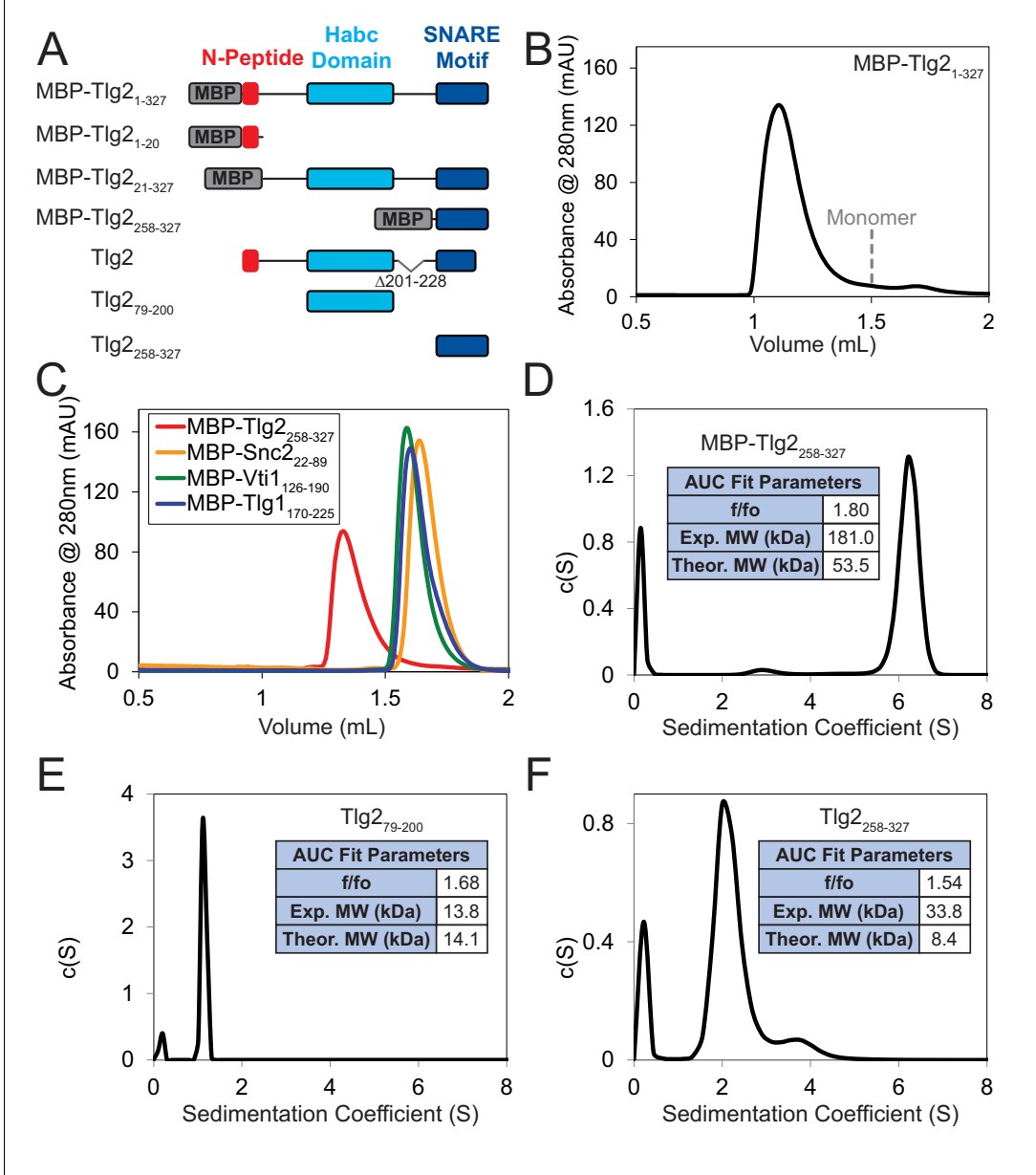

**Figure 5.** Homo-oligomerization of the Tlg2 SNARE motif. (**A**) Schematic of the Tlg2 constructs used. (**B**) Size exclusion chromatography of MBP-Tlg2$_{1-327}$. The predicted position for a monomer, based on size standards, is indicated. (**C**) Size-exclusion chromatography of the MBP-tagged *C. thermophilum* SNARE motifs of Tlg2 (Qa-SNARE), Snc2 (R-SNARE), Vti1 (Qb-SNARE), and Tlg1 (Qc-SNARE). (**D–F**) Sedimentation velocity analytical ultracentrifugation (AUC) and derived parameters (insets). For MBP-Tlg2$_{258-327}$ (panel D), the experimental molecular weight (181 kDa) falls between those expected for a trimer (161 kDa) and a tetramer (214 kDa); for untagged Tlg2$_{258-327}$ (panel F), the experimental molecular weight (33.8 kDa) is in excellent agreement with that expected for a tetramer (33.6 kDa). The Habc domain (Tlg2$_{79-200}$) sediments as a monomer (panel E). For data and fits, see *Figure 5—figure supplement 1*.

The online version of this article includes the following figure supplement(s) for figure 5:

**Figure supplement 1.** Sedimentation velocity analytical ultracentrifugation data and fits.

Vps45–Tlg2 structure, Tlg2 is open but nevertheless protected from oligomerization. We propose that this protection arises from binding of the SM protein to layers 0 to +4 of the SNARE motif (*Figure 2B*). The other principal interaction between Vps45 and Tlg2, involving the Tlg2 N-peptide, increases the stability of the Vps45–Tlg2 complex but would not seem capable of preventing SNARE

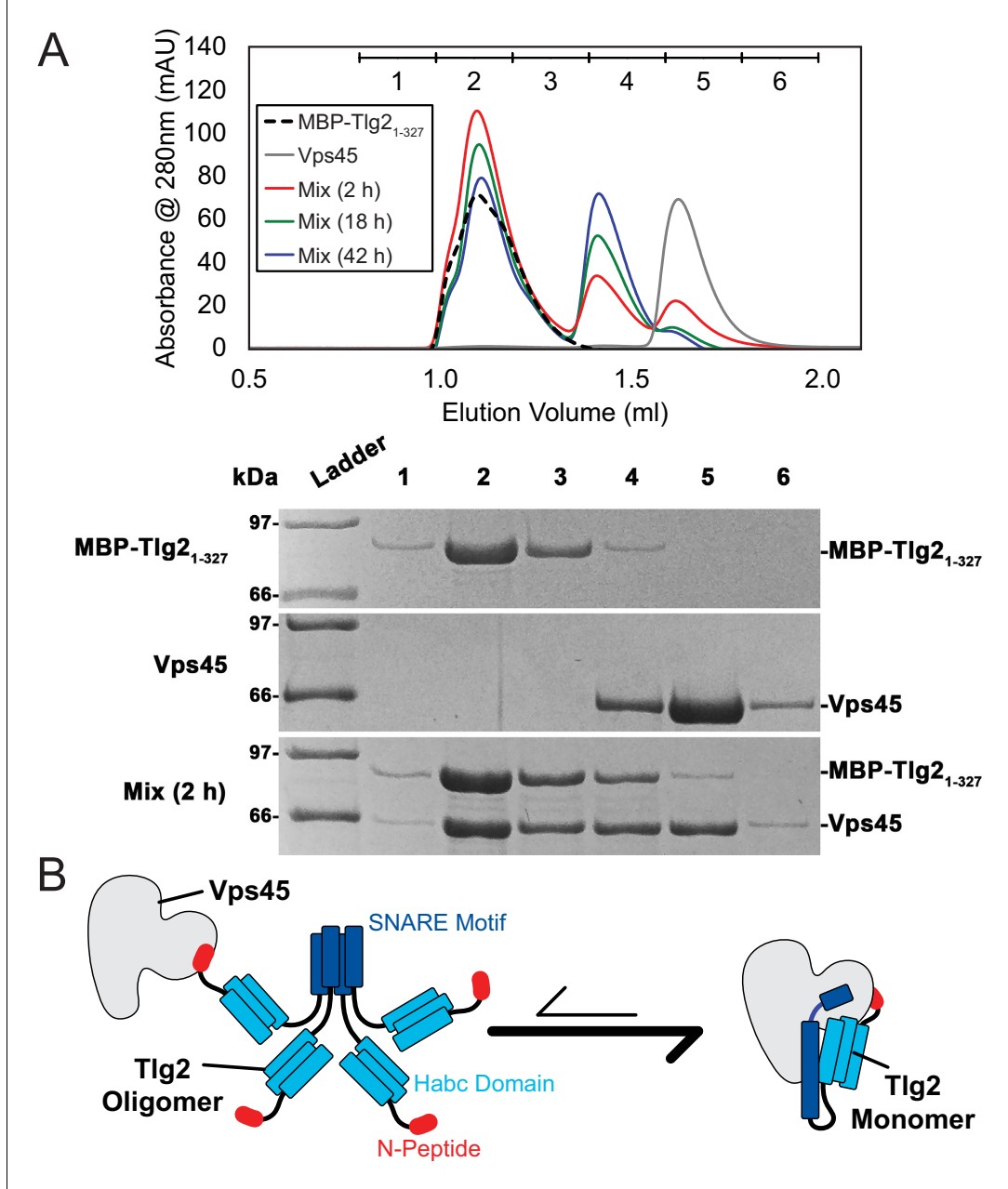

**Figure 6.** Vps45 rescues Tlg2 from homo-oligomers. (**A**) Size-exclusion chromatography (top) of MBP-Tlg2$_{1-327}$, Vps45, and a 1:1 mixture incubated at 20°C for 2, 18, or 42 hr. The indicated fractions were analyzed by SDS-PAGE (bottom). (**B**) Schematic representation of Vps45-mediated rescue of Tlg2 from homo-oligomers.

The online version of this article includes the following figure supplement(s) for figure 6:

**Figure supplement 1.** The Tlg2 N-Peptide is required for Vps45 rescue.

motif self-association. The Tlg2 N-peptide nevertheless plays a key role, by equipping the Tlg2 tetramers with exposed, high-affinity (27 nM for the mammalian orthologues [*Burkhardt et al., 2008*]) handles for Vps45 to grab onto (*Figure 6B*). Taken together, our results suggest that Vps45 can rescue Tlg2 from off-pathway oligomers and hold it in an open, non-oligomerization-prone state in preparation for SNARE complex assembly (*Figure 7*).

Template complexes containing half-zippered Qa- and R-SNAREs bound to the cognate SM protein are essential intermediates in SM-catalyzed SNARE assembly (*Baker et al., 2015*; *Jiao et al.,*

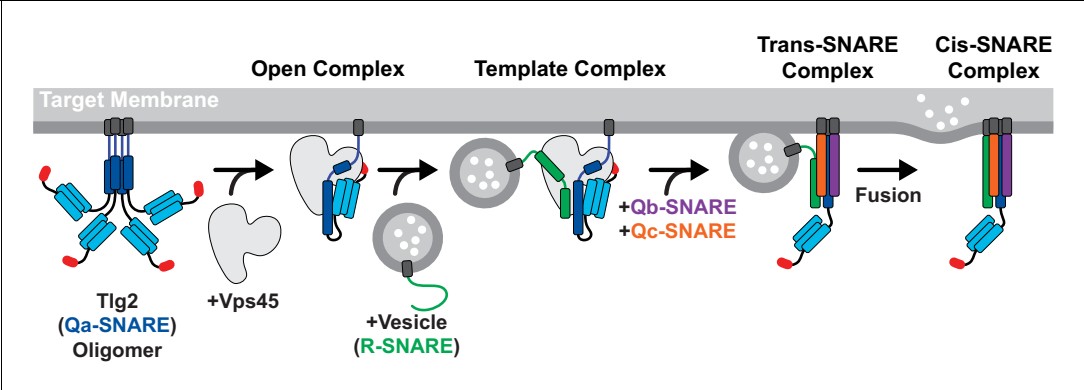

**Figure 7.** Model for Vps45-mediated SNARE assembly. Proposed pathway depicting Tlg2 rescue, vesicle docking accompanied by template complex formation, Qb/Qc-SNARE binding, and membrane fusion.

*2018*). Yeast Vps45 was reported to bind the R-SNARE Snc2, but the relevance of this observation to the potential formation of a template complex is uncertain since Tlg2 appeared to displace the Vps45-bound Snc2 (*Carpp et al., 2006*). Nonetheless, the close structural resemblance between the unfurled helical hairpins of Tlg2-bound Vps45, Vam3-bound Vps33, and Nyv1-bound Vps33, together with the ability of Vps33, Nyv1, and Vam3 to form a ternary complex (*Baker et al., 2015*), strongly suggests that Tlg2-bound Vps45 is primed for R-SNARE binding (*Figure 4C*). This binding need not be high affinity in order to promote SNARE assembly; indeed, Munc18 binding to the R-SNARE VAMP2/synaptobrevin is weak (*Sitarska et al., 2017*) but nonetheless critical for forming the template complex (*Jiao et al., 2018*). It will be important in future work to develop biochemical and single-molecule methods for elucidating the pathway and energetics of SNARE assembly in the presence of Vps45.

The stringent regulation of neurotransmitter release requires the control of synaptic SNARE complex formation by Munc18, which clamps Stx in an inactive closed conformation, and Munc13-1, which mediates Stx opening (*Brunger et al., 2018*; *Rizo, 2018*). Our Vps45–Tlg2 structure reveals a new mode of interaction in which the Qa-SNARE is held open rather than clamped shut. It is possible that this new mode is the rule, with the Munc18–Stx mode being the exception. It is furthermore attractive to speculate that the task of Munc13-1 is to convert the Munc18–Stx complex into a Vps45–Tlg2-like conformation. Notably, a principal difference between closed Stx and open Tlg2 is the linker between the Habc domain and the SNARE motif, which is folded in the first case and unfolded in the second (*Figure 3A*). Munc13-1, by binding to the Munc18–Stx complex and destabilizing the Stx linker (*Lai et al., 2017*; *Wang et al., 2017*; *Wang et al., 2020*), may thereby promote the conformational switch that allows Stx to open, the domain 3a helical hairpin to unfurl, and the template complex to form.

## Materials and methods

### Recombinant protein production

*C. thermophilum* gene sequences were identified by homology with multiple fungal species (BLAST) yielding Vps45 (XP_006692860.1), Tlg2 (XP_006697074.1), Snc2 (XP_006691992.1), Vti1 (XP_006696366.1), and Tlg1 (XP_006693250.1). Coding sequences were amplified from synthetic codon-optimized genes (Genewiz) and cloned into pQLink bacterial expression plasmids (*Baker et al., 2015*; *Scheich et al., 2007*). All SNARE protein constructs possessed N-terminal heptahistidine (His$_7$) and MBP tags followed by a tobacco etch virus (TEV) protease cleavage site for tag removal, with the exception of Tlg2$_{79-200}$ which carried N-terminal His$_7$ and SUMO tags and an Ulp1 cleavage site for tag removal. Vps45 constructs possessed an N-terminal His$_7$ tag for individual expression, or a C-terminal His$_7$ tag for coexpression with Tlg2. The Vps45–Tlg2 coexpression plasmid was generated by first deleting the His$_7$-MBP tag from Tlg2 and then combining the two pQLink

plasmids as described (*Scheich et al., 2007*). Mutations were introduced using QuikChange mutagenesis (Agilent). Tlg2 sub-fragment constructs were generated by introducing stop codons or by using PCR to amplify the desired fragment for sub-cloning.

Native and selenomethionine (SeMet) Vps45 was overproduced using BL21 Rosetta (Novagen) grown in, respectively, LB or M9 minimal media supplemented with 60 mg/L SeMet (Sigma). When the cells reached an $OD_{600}$ of approximately 0.6, isopropyl β-D-thiogalactopyranoside (IPTG) was added to a final concentration of 1 mM, after which the cells were grown for 18 hr at 25°C. MBP-$Tlg2_{1-20}$, MBP-$Tlg2_{21-327}$, MBP-$Tlg2_{258-327}$, SUMO-$Tlg2_{79-200}$, MBP-$Snc2_{22-89}$, MBP-$Tlg1_{170-225}$, and MBP-$Vti1_{126-190}$ were overproduced in a similar manner, but in BL21-Codon Plus (Agilent) cells and with growth following induction at 18°C. Cell pellets were resuspended in lysis buffer (20 mM Tris-HCl pH 8.0, 150 mM NaCl, 5 mM ß-mercaptoethanol) supplemented with 1 mM phenylmethylsulfonyl fluoride and 10 µg/mL DNase (Roche). The resuspension was processed with an Emulsiflex-C5 homogenizer (Avestin). All subsequent steps were performed on ice or at 4°C. The cell lysate was clarified by centrifugation at 30,000 g and fractionated using His60 Ni Superflow Resin (ClonTech). The resin was washed using wash buffer (20 mM Tris-HCl pH 8.0, 100 mM NaCl, 20 mM imidazole, 5 mM ß-mercaptoethanol), eluted in wash buffer containing 300 mM imidazole, and further purified using a Superdex 200 HR 16/60 size exclusion column (GE Healthcare) in gel filtration buffer (20 mM Tris-HCl pH 8.0, 150 mM NaCl, 5 mM dithiothreitol (DTT)). MBP-$Tlg2_{1-327}$ and the co-expressed Vps45–Tlg2 complexes were generated in a similar manner with the following modifications: protein was overexpressed in BL21-Codon Plus (Agilent); after IPTG addition the cells were grown at 16°C; and all buffers contained 5% glycerol and, in place of Tris-HCl, 25 mM HEPES pH 8.0. Cells were lysed, loaded onto His60 Ni Superflow Resin, and washed as previously described. However, the resin was then washed with low salt (50 mM NaCl) wash buffer and eluted in low salt wash buffer containing 400 mM imidazole, followed by anion exchange chromatography (MonoQ 10/100; GE Healthcare) using a gradient from 50 mM to 500 mM NaCl. Untagged $Tlg2_{79-200}$ and $Tlg2_{258-327}$ were produced by cleaving the SUMO- and MBP-tagged fusion proteins with Ulp1 or TEV protease, respectively, following Ni affinity chromatography. Cleavage reactions were allowed to proceed overnight at 4°C while dialyzing into elution buffer containing 20 mM imidazole and 50 mM NaCl. The cleaved tags were removed using His60 Ni Superflow resin and the proteins were further purified using anion exchange and size exclusion chromatography as described above. For the production of SeMet-substituted Vps45–$Tlg2_{L258M,I272M}$, protein expression was improved by growing the cells in M63 instead of M9 minimal media. Following size exclusion chromatography, all proteins were concentrated, snap frozen in liquid nitrogen, and stored at −80°C. Protein concentration was measured by absorbance at 280 nm.

## Crystallization, data collection, and refinement

Vps45 crystals were grown at 20°C using the sitting drop vapor diffusion method, with a 1:1 (v/v) mixture of protein at 10 mg/ml and precipitant solution (0.2 M potassium bromide, 0.2 M potassium thiocyanate, 0.1 M sodium acetate pH 6.0, 3% (w/v) poly-γ-glutamic acid (PGA), 5% (w/v) polyethylene glycol (PEG) 3350). The final drop volume was 1 µl, brought to equilibrium with 500 µl precipitant solution. SeMet crystals grew to full size within two weeks and native crystals within three weeks. Native and SeMet single wavelength anomalous diffraction data were collected at the Cornell High Energy Synchrotron Source at beamline F1.

The Vps45 structure was determined by experimental phasing at 2.6 Å resolution using the single wavelength anomalous dispersion method based on diffraction at the Se K edge (λ = 0.9782 Å). The selenium substructure was determined using the program SHELXD (*Sheldrick, 2008*) and phases were calculated with SHARP (*Vonrhein et al., 2011*) based on these sites. The electron density map was solvent-flattened using SOLOMON (*Abrahams and Leslie, 1996*), and the structure built using BUCCANEER (*Cowtan, 1998*) and Coot (*Emsley et al., 2010*) and refined against higher-resolution native data using PHENIX REFINE (*Liebschner et al., 2019*).

All Vps45–Tlg2 complexes were crystallized using hanging drop vapor diffusion at 20°C, mixing 1 µl protein with 1 µl well buffer solution. Vps45–$Tlg2_{1-310}$ crystals were grown using 4 mg/ml protein and well buffer consisting of 0.125 M potassium citrate, 15% (w/v) PEG 3350, 15% (v/v) glycerol, and 1 mM DTT. Crystals were improved by streak seeding with previously grown Vps45–$Tlg2_{1-310}$ crystals. Vps45–Tlg2 (i.e., Vps45–$Tlg2_{1-310, \Delta201-228}$) crystals were grown using 2.5 mg/ml protein and well buffer consisting of 0.1 M HEPES pH 8.0, 0.1 M NaCl, 11% (v/v) 2-propanol, and 5 mM

TCEP. Vps45–Tlg2$_{V306D,F335R}$ crystals were grown using 4 mg/ml protein solution and well buffer consisting of 0.2 M potassium citrate, 14% (w/v) PEG 3350, and 5 mM TCEP. SeMet-labeled Vps45–Tlg2$_{L258M,I272M}$ crystals were grown using 5 mg/ml protein solution and well buffer consisting of 0.1 M HEPES pH 7.5, 0.2 M NaCl, 10% (v/v) 2-propanol, and 5 mM TCEP. Crystals were improved using streak seeding with Vps45–Tlg2 crystals. Diamond-shaped crystals grew to full size within ~3 days. Crystals were cryoprotected using a 1:1 mixture of well buffer supplemented with 30% (v/v) glycerol (for Vps45–Tlg2$_{1-310}$ and Vps45–Tlg2$_{V306D,F335R}$) or 30% (v/v) glycerol plus 10% (v/v) 2-propanol (for Vps45–Tlg2 and SeMet-labeled Vps45–Tlg2$_{L258M,I272M}$) and then frozen in liquid nitrogen. Data were collected at the National Synchrotron Light Source II (NSLSII) FMX and AMX beamlines.

The structures of the Vps45–Tlg2 complexes were determined by molecular replacement from the Vps45 monomer structure using the program PHASER (*McCoy et al., 2007*). Complexes grew in two different crystal forms: a form with a single complex in the asymmetric unit in space group P2$_1$22$_1$ with typical cell dimensions of a = 58.4 Å, b = 89.4 Å, c = 209.1 Å, and a form with two complexes in the asymmetric unit in space group P2$_1$2$_1$2$_1$ with typical cell dimensions of a = 58.7 Å, b = 180.1 Å, c = 202.1 Å. Vps45–Tlg2 grew in both crystal forms but Vps45–Tlg2$_{1-310}$ only grew in the P2$_1$22$_1$ crystal form. Difference density for the N-terminal peptide, Habc domain, and SNARE helices were visible in the difference map in both cases. Crystal packing is similar in these two crystal forms, with the SNARE helix and domain 3a of Vps45 packing against symmetry-related instances of themselves to stabilize the lattice. Models of the Vps45–Tlg2 complexes were built with Coot and refined with PHENIX.REFINE. At intermediate steps in structure determination we utilized data processed by STARANISO (RRID:SCR_018362) with an anisotropic resolution cutoff to improve map interpretability, but results quoted in *Table 1* are refined against data with an isotropic resolution cutoff. Non-crystallographic symmetry restraints were used where available. The useful resolution limits of the data were estimated using the paired refinement technique (*Diederichs and Karplus, 1997*). Sequence interpretation for Tlg2 was made based on 2Fo-Fc and Fo-Fc electron density and validated by the position of selenium atoms in SAD data collected on SeMet-labeled Vps45–Tlg2$_{L258M,I272M}$ crystals. The selenium sites corresponding to L258M and I272M mutations confirmed the sequence register in the SNARE helix, and selenium sites corresponding to Met 1, Met 153 and Met 170 confirmed the assignment in the N-peptide and Habc domains, with Met 302 confirming the conserved H3c helix assignment. Electron density for Tlg2 is in general less well-resolved than that for Vps45, and especially so at the ends of the Habc helices at the C-terminal end of the bundle (distal to the Vps45:Habc binding site), likely reflecting some static disorder within the crystal. Nevertheless, the similarity with the Habc helices of Stx is striking (rmsd = 1.3 Å for 91 Cα atoms).

## Binding/rescue assays

Vps45 and MBP-Tlg2$_{1-327}$ were mixed at a final concentration of 30 µM each in gel filtration buffer and incubated at 20°C. At the specified times, aliquots were removed and snap frozen in liquid nitrogen followed by storage at −80°C. Prior to loading, samples were thawed rapidly in a room temperature water bath and any large aggregates were removed via centrifugation at 14,000 g for 10 min at 4°C. Samples were then loaded on an S200 Increase 3.2/300 gel filtration column (GE Healthcare) pre-equilibrated with gel filtration buffer. Fractions were resolved using 12.5% SDS-PAGE gels. All other binding experiments were performed using 50 µM protein concentration(s) in 20 mM Tris-HCl, pH 8.0, 150 mM NaCl, and 1 mM DTT, with a 1 hr incubation at 20°C followed by centrifugation to remove aggregates and loading onto the size exclusion column.

## Analytical ultracentrifugation

MBP-Tlg2$_{258-327}$, Tlg2$_{79-200}$, and Tlg2$_{258-327}$ were diluted to final concentrations of 9, 72, and 168 µM, respectively, in gel filtration buffer. Samples were centrifuged at 14,000 g for 10 min at 10°C, loaded into two-sector cells with quartz windows, and placed in an Optima analytical centrifuge (Beckman) with an An-50Ti rotor pre-equilibrated at 20°C. Absorbance scans at 280 nm were collected at ~1 min intervals while spinning at 42,000 rpm. Continuous sedimentation coefficient c(S) plots and frictional ratios (f/f$_o$) were generated using SEDFIT by fitting the Lamm equation to the absorbance boundaries (*Schuck, 2000*). Buffer density (r, 1.0064 g/cm$^3$) and viscosity (h, 0.01 poise) were calculated using SEDNTERP (*Laue et al., 1992*).

## Acknowledgements

We thank Mary Munson, Jose Rizo, Venu Vandavasi, Yongli Zhang, and members of the Hughson laboratory past and present for helpful advice and discussion. The Princeton Biophysics and Macromolecular Crystallography core facilities provided essential assistance with analytical ultracentrifugation and X-ray crystallography, respectively. This work was supported by National Institutes of Health (NIH) grants T32GM007388 (GRS) and R01GM071574 (FMH). This research used the AMX and FMX beamlines of the National Synchrotron Light Source II, a United States Department of Energy (DOE) Office of Science User Facility operated for the DOE Office of Science by Brookhaven National Laboratory under Contract No. DE-SC0012704. The Center for BioMolecular Structure (CBMS) is primarily supported by the NIH, National Institute of General Medical Sciences (NIGMS) through a Center Core P30 Grant (P30GM133893), and by the DOE Office of Biological and Environmental Research (KP1605010).This work is based upon research conducted at the Cornell High Energy Synchrotron Source (CHESS), which is supported by the National Science Foundation (NSF) and the NIH/NIGMS under NSF award DMR-1829070, using the Macromolecular Diffraction at CHESS (MacCHESS) facility, which is supported by NIH/NIGMS award GM-124166.

## Additional information

### Funding

| Funder | Grant reference number | Author |
|---|---|---|
| National Institutes of Health | R01GM071574 | Frederick M Hughson |
| National Institutes of Health | T32GM007388 | Gregory R Shimamura |

The funders had no role in study design, data collection and interpretation, or the decision to submit the work for publication.

### Author contributions

Travis J Eisemann, Conceptualization, Formal analysis, Investigation, Writing - original draft, Writing - review and editing; Frederick Allen, Kelly Lau, Gregory R Shimamura, Conceptualization, Investigation, Writing - review and editing; Philip D Jeffrey, Conceptualization, Data curation, Formal analysis, Validation, Investigation, Writing - review and editing; Frederick M Hughson, Conceptualization, Formal analysis, Supervision, Funding acquisition, Writing - original draft, Project administration, Writing - review and editing

### Author ORCIDs

Travis J Eisemann https://orcid.org/0000-0003-3602-2677
Frederick Allen https://orcid.org/0000-0002-2969-8137
Gregory R Shimamura http://orcid.org/0000-0003-2104-5518
Frederick M Hughson https://orcid.org/0000-0002-4057-0281

### Decision letter and Author response

Decision letter https://doi.org/10.7554/eLife.60724.sa1
Author response https://doi.org/10.7554/eLife.60724.sa2

## Additional files

### Supplementary files

• Transparent reporting form

### Data availability

Diffraction data have been deposited in the PDB under the accession codes 6XJL, 6XMD, and 6XM1.

The following datasets were generated:

| Author(s) | Year | Dataset title | Dataset URL | Database and Identifier |
|---|---|---|---|---|
| Jeffrey PD, Shimamura GS, Allen F, Hughson FM | 2020 | Structure of the SM protein Vps45 | https://www.rcsb.org/structure/6XJL | RCSB Protein Data Bank, 6XJL |
| Jeffrey PD, Eisemann TJ, Hughson FM | 2020 | SM Protein Vps45 in Complex with Qa SNARE Tlg2 (1-310) | https://www.rcsb.org/structure/6XMD | RCSB Protein Data Bank, 6XMD |
| Jeffrey PD, Eisemann TJ, Hughson FM | 2020 | SM Protein Vps45 in Complex with Qa SNARE Tlg2 | https://www.rcsb.org/structure/6XM1 | RCSB Protein Data Bank, 6XM1 |

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
