## [Decision Letter]

**Acceptance summary:**

In this paper, the first structure of an SM complex has been determined where the cognate syntaxin (Tlg2) is bound in an open conformation to the SM protein Vps45 in contrast to the closed confirmation that was found in the structure of the syntaxin-Munc18 complex. This structure suggests that ternary SNARE complex formation can occur starting from the Vps45/Tlg2 complex without assistance of a catalyst such as Munc13. These findings have implications how SM proteins have evolved.

**Decision letter after peer review:**

Thank you for submitting your article "The Sec1/Munc18 protein Vps45 holds the Qa-SNARE Tlg2 in an open conformation" for consideration by *eLife*. Your article has been reviewed by three peer reviewers, one of whom is a member of our Board of Reviewing Editors, and the evaluation has been overseen by Suzanne Pfeffer as the Senior Editor. The following individual involved in review of your submission has agreed to reveal their identity: Cong Ma, Christopher Fromme.

The reviewers have discussed the reviews with one another and the Reviewing Editor has drafted this decision to help you prepare a revised submission.

Summary:

In this paper, the first structure of an SM complex has been determined where the cognate syntaxin (Tlg2) is bound in an open conformation to the SM protein Vps45 in contrast to the closed confirmation that was found in the structure of the syntaxin-Munc18 complex. A loop that is critical for the Munc13 catalyzed opening of the syntaxin-Munc18 complex is 'unfurled', i.e., it extends out rather than folds back. This structure suggests that ternary SNARE complex formation can occur starting from the Vps45 / Tlg2 complex without assistance of a catalyst such as Munc13. Moreover, Vps45 is capable of dissociating Tlg2 oligomers by formation of 1:1 complexes, suggesting that Vps45 induces a template conformation of Tlg2 for cognate SNARE binding. These findings have implications how SM proteins have evolved.

Essential revisions:

1) The open conformation of Tlg2 bound to Vps45 renders it easier to assemble into the SNARE complex with its partner SNAREs Vti1, Tlg1, and Snc1/2. Please provide additional biochemical data to elucidate the difference of the assembly rate with or without Vps45. In addition, the activity of Vps45-Tlg1-310 and Vps45-Tlg1-310, delete Δ201-228 in SNARE complex assembly should be carefully examined and compared, because deletion of residues 201-228 might affect the linker structure of Tlg1.

2) The authors write: "Unfortunately, we were unable to identify in vitro conditions under which *C. thermophilum* Tlg2 with or without Vps45, assembles into SNARE complexes." Were different fragments of the various SNARE motifs tried? It seems very surprising that a SNARE complex cannot be formed in vitro. Please comment.

3) The MBP-Tlg2[1-327] chimera forms oligomers as suggested by size exclusion chromatography and analytical ultracentrifugation. As an optional follow-up experiment it would be instructive to show that tag free Tlg2 forms trimers or tetramers on its own. If tag-free Tlg2 is poorly behaved, perhaps a different tag could be tried.

4) The authors state "based on the crystal structure, the Vps45-Tlg2 complex appears to be primed to bind an R-SNARE and, as demonstrated for SM proteins of the Munc18/Sec1 and Vps33 families (Jiao et al., 2018), to catalyze SNARE assembly". Is it possible to generate a structural model by superimposing the Vps33/R-SNARE structure on the Vps45/Qa-SNARE structure?

5) Is it possible that Tlg2 never adopts a canonical "closed" structure and this is why it oligomerizes?

6) Similarly, is the Tlg2 Habc domain known to be autoinhibitory? Has this ever been tested for Tlg2?

---

## [Author Response]

Essential revisions:1) The open conformation of Tlg2 bound to Vps45 renders it easier to assemble into the SNARE complex with its partner SNAREs Vti1, Tlg1, and Snc1/2. Please provide additional biochemical data to elucidate the difference of the assembly rate with or without Vps45. In addition, the activity of Vps45-Tlg1-310 and Vps45-Tlg1-310, delete Δ201-228 in SNARE complex assembly should be carefully examined and compared, because deletion of residues 201-228 might affect the linker structure of Tlg1.

Unfortunately, despite extensive efforts, we have been unable to observe assembly of *C. thermophilum* Tlg2 into SNARE complexes. This is true for Tlg2 by itself, which one might attribute to the oligomerization of Tlg2, but also for Tlg2:Vps45 complexes (see Point 2). Therefore we are currently unable to compare assembly rates. It does however seem highly unlikely that deleting residues 201-228 would affect key properties of Tlg2, because this region is missing from the otherwise nearly identical *Chaetomium globosum* Tlg2 (subsection “Crystal Structure of Vps45–Tlg2” and Figure 1—figure supplement 1B).

2) The authors write: "Unfortunately, we were unable to identify in vitro conditions under which C. thermophilum Tlg2 with or without Vps45, assembles into SNARE complexes." Were different fragments of the various SNARE motifs tried? It seems very surprising that a SNARE complex cannot be formed in vitro. Please comment.

We tried many different Tlg2 constructs, thinking that perhaps we could mitigate its self-association without abolishing its ability to bind its partners, but this approach was unsuccessful. It remains possible that varying the other SNARE constructs might resolve the problem, and we appreciate the reviewer’s suggestion. At this juncture, however, we feel that it is beyond the scope of the current manuscript to undertake further exploratory experiments. As an alternative approach, we are initiating single-molecule optical tweezers studies in collaboration with Yongli Zhang’s lab at Yale. We write: “It will be important in future work to develop biochemical and single-molecule methods for elucidating the pathway and energetics of SNARE assembly in the presence of Vps45.”

3) The MBP-Tlg2[1-327] chimera forms oligomers as suggested by size exclusion chromatography and analytical ultracentrifugation. As an optional follow-up experiment it would be instructive to show that tag free Tlg2 forms trimers or tetramers on its own. If tag-free Tlg2 is poorly behaved, perhaps a different tag could be tried.

In response to the reviewer’s suggestion, we conducted additional analytical ultracentrifugation experiments using tag-free Tlg2 constructs representing the Habc domain and SNARE motif. The results are presented in new Figure 5E and F. While the untagged Habc domain is monomeric, the untagged SNARE motif is tetrameric. These new results solidify our conclusion that Tlg2, like Stx, tetramerizes via its SNARE motif.

4) The authors state "based on the crystal structure, the Vps45-Tlg2 complex appears to be primed to bind an R-SNARE and, as demonstrated for SM proteins of the Munc18/Sec1 and Vps33 families (Jiao et al., 2018), to catalyze SNARE assembly". Is it possible to generate a structural model by superimposing the Vps33/R-SNARE structure on the Vps45/Qa-SNARE structure?

It follows from the close resemblance between the domain 3a helical hairpins of Vps33 and Vps45 (Figure 3B) that their R-SNARE binding grooves are similar. To illustrate the putative ternary complex, we superimposed the Vps33/R-SNARE structure on the Vps45/Qa-SNARE structure as requested (new Figure 3C). We note that we and others are actively attempting to generate actual structures of template complexes.

5) Is it possible that Tlg2 never adopts a canonical "closed" structure and this is why it oligomerizes?

This is indeed possible. In fact a seminal paper, Dulubova et al., 2002, presented NMR studies of yeast Tlg2 that argued strongly against a stable closed conformation. The caveat was that, in order to mitigate Tlg2 oligomerization, it was necessary to remove the C-terminal half of the SNARE motif; nevertheless, the authors retained that part of the SNARE motif that was involved in forming the closed conformation in Stx and Sso1. Later biochemical analyses consistent with the possibility of a closed conformation were presented by Furgason et al., 2009, and Struthers et al., 2009, but the evidence was indirect. We summarize these conflicting results by writing: “There have been conflicting reports regarding the tendency of yeast Tlg2 to oligomerize and/or form a closed conformation (Dulubova et al., 2002; Furgason et al., 2009; Struthers et al., 2009).” While this is admittedly rather terse, we feel that – given the propensity of *C. thermophilum* Tlg2 to form presumably-open tetramers – we don’t presently have anything new to add to the debate. In the future, we hope that single-molecule experiments (see response to Point 2 above) will help to clarify the existence and significance of the closed conformation in relation to SNARE assembly.

6) Similarly, is the Tlg2 Habc domain known to be autoinhibitory? Has this ever been tested for Tlg2?

Consistent with this possibility, Bryant and James, 2001, found that deletion of the first 230 residues of Tlg2, including the Habc domain, could rescue SNARE complex formation in yeast lacking Vps45. Struthers et al., 2009, presented a pull-down experiment in which maximally efficient SNARE assembly by immobilized yeast Tlg2 seemed to require either the deletion of the Habc domain or the presence of Vps45. By contrast, as noted above, NMR revealed no evidence for an autoinhibitory closed conformation (Dulubova et al., 2002). Once again, we are hopeful that single-molecule experiments will ultimately contribute to the resolution of these apparently conflicting results.